# Hidden Partner of Immunity: Microbiome as an Innovative Companion in Immunotherapy

**DOI:** 10.3390/ijms26020856

**Published:** 2025-01-20

**Authors:** Pyoseung Kim, Sunggeun Joe, Heeyoung Kim, Hyejeong Jeong, Sunghwan Park, Jihwan Song, Wondong Kim, Yong Gu Lee

**Affiliations:** College of Pharmacy, Institute of Pharmaceutical Science and Technology, Hanyang University, Ansan 15588, Gyeonggi-do, Republic of Korea; rubykp@hanyang.ac.kr (P.K.); jiy7991@hanyang.ac.kr (S.J.); hypsim12@hanyang.ac.kr (H.K.); wjdgpwjd30@hanyang.ac.kr (H.J.); jason630@hanyang.ac.kr (S.P.); mejhjh@hanyang.ac.kr (J.S.)

**Keywords:** immunotherapy, CAR-T cell therapy, mesenchymal stem cells, antibody therapy, microbiome, genetic engineered microbiome

## Abstract

Recent studies have highlighted that the microbiome is the essential factor that can modulate the clinical activity of immunotherapy. However, the role of the microbiome varies significantly across different immunotherapies, suggesting that it is critical to understand the precise function of the microbiome in each type of immunotherapy. While many previous studies primarily focus on summarizing the role of the microbiome in immune checkpoint inhibitors, we seek to explore a novel aspect of the microbiome in other immunotherapies such as mesenchymal stem cell therapy, chimeric antigen receptor T cell therapy, and antibodies-based therapy (e.g., adalimumab, infliximab, bevacizumab, denosumab, etc.) which are rarely summarized in previous reviews. Moreover, we highlight innovative strategies for utilizing microbiome and microbial metabolites to enhance the clinical response of immunotherapy. Collectively, we believe that our manuscript will provide novel insights and innovative approaches to the researchers, which could drive the development of the next generation of personalized therapeutic interventions using microbiomes.

## 1. Introduction

A microbiome is an intricate ecosystem of over a trillion organisms inhabiting various organs of the human body, including the gastrointestinal tract. The microbiome plays a critical role in digestion, immune homeostasis, and metabolic activities by supporting nutrient absorption, synthesizing essential vitamins, and regulating inflammation in the immune system.

Maintaining gut microbiome diversity is critical in optimizing their biological activity. Therefore, dysbiosis occurs when the balance in the gut microbiome is disturbed, causing several chronic diseases such as inflammatory bowel disease, obesity, diabetes, and neurodegenerative diseases. For example, McCoy et al. reported that individuals with an increased number of *Fusobacterium* in their gut microbiome can develop colorectal cancer and adenomas [1,2]. Furthermore, dysbiosis within the gut microbiome is associated with developing hepatocellular carcinoma by facilitating chronic inflammation through toll-like receptor (TLR) signal pathways. Dysbiosis leads to immunological disorders such as irritable bowel syndrome (IBS) [3,4], rheumatoid arthritis (RA) [5], and systemic lupus erythematosus (SLE) [6], among others. Scher et al. identified the enrichment of *Prevotella copri* in patients with new-onset RA, linking it to increased pro-inflammatory cytokine production and joint inflammation [5]. Zegarra-Ruiz et al. discovered that dietary supplementation with a commensal *Lactobacillus* strain reduced autoimmune markers and kidney damage in lupus-prone mice by modulating TLR7 pathways [6]. In addition, the gut microbiome is associated with neurological and respiratory diseases in childhood and infancy. Liu et al. revealed that antibiotics can alter the gut microbiome balance in infants without a fully matured microbiome system, leading to the development of neurological disorders such as autism spectrum disorder (ASD) [7]. Stokholm et al. investigated the relationship between the composition of the gut microbiome during the first year of life and the risk of developing asthma at 5 years old in 690 children from the Copenhagen Pediatric Asthma Prospective Study (COPSAC 2010) cohort [8]. In this study, the researchers discovered that the low abundance of a specific bacterial genera (e.g., *Faecalibacterium*, *Bifidobacterium*) and the high abundance of *Veillonella* are strongly associated with an elevated risk of asthma. Altogether, these studies strongly suggest that the microbiome has a significant role in host health and disease development.

Immunotherapy is a novel treatment that utilizes the immune system to resist various diseases, including cancer and autoimmune diseases. Immune checkpoint inhibitors (ICIs), a subtype of immunotherapy, enhance the body’s immune response to cancer by targeting immune checkpoints. Immune checkpoints are regulatory pathways that preserve self-tolerance and prevent autoimmunity. These checkpoints are exploited by tumors to evade immune surveillance. ICIs inhibit these checkpoints, effectively allowing immune cells, particularly T cells, to generate a robust antitumor response [9,10]. The microbiome’s interaction and immunotherapy has been a growing interest in cancer therapeutics. Several studies have shown that some gut bacteria can enhance the efficacy of immunotherapy. Vétizou et al. discovered an association between the effectiveness of Cytotoxic T-lymphocyte-associated protein 4 (CTLA-4) blockade therapy and the genus *Bacteroides* [11]. Moreover, Sivan et al. found a correlation between the effectiveness of Programmed death-ligand 1 (PD-L1) blockade therapy and *Bifidobacterium* [12]. Smith et al. observed that the efficacy and toxicity of Chimeric Antigen Receptor (CAR)-T cell therapy might be affected by particular gut microbiome compositions in varying aspects of immunotherapy, such as cell therapy [13]. For example, the presence of strains such as *Ruminococcaceae* and *Eubacterium halii* was associated with an improved CAR-T cell therapy response. These strains promote immune regulation by producing monoclonal fatty acids. In contrast, increased Bacteroidetes genus (specifically *B. thetaiotaomicron* and *B.ovatus*) and *Veillonella parvula* were associated with poor response to CAR-T cell therapy. Such interactions could provide better insights toward developing a better strategy for increasing the effectiveness of immunotherapy while decreasing its related adverse effects.

Evaluating the microbiome’s effect in modulating the clinical response of immunotherapy to develop effective therapeutic strategies is crucial, considering the significant association of the microbiome with immunotherapy. Therefore, in this review, we aimed to review recent advances in studying the relationship between microbiome and immunotherapy. We will primarily introduce the interaction between the microbiome and different forms of immunotherapies such as CAR-T therapy, mesenchymal stem cells (MSC) therapy, and antibody therapy. We exclude ICI, as it is widely studied and its effectiveness associated with the microbiome is well summarized [14,15]. Furthermore, we will conclude this review by discussing how microbiomes can be used to enhance the efficacy of immunotherapy. In this paper, we summarize existing studies to elucidate the relationship between the microbiome and immunotherapy. Key findings are organized and visually represented in Figure 1, which overviews the main topics discussed in this review.

## 2. The Effect of Microbiome in Immunotherapy

### 2.1. CAR-T Cell Therapy

CAR-T cell therapy is a novel immunotherapy that uses genetically modified patient’s immune cells to target cancer [16,17]. CAR-T cells can specifically recognize cancer cells and induce activation signals with CAR-containing tumor antigen binding and signaling domains [18]. Currently, seven US Food and Drug Administration (FDA)-approved CAR-T cell therapies (Kymriah, Yescarta, Tecartus, Breyanzi, Aucatzyl, Abecma, and Carvykti [19,20]) are used for the treatment of hematological malignancy in clinics. CAR-T therapies demonstrated their substantial antitumor activities; however, 60% of patients showed no clinical response or experienced tumor relapse following CAR-T treatment [21]. In addition, CAR-T therapy can cause several adverse effects, comprising cytokine release syndrome (CRS) and/or immune effector cell-associated neurotoxicity syndrome (ICANS) [22,23].

Recent studies have identified that the gut microbiome influences the efficacy and toxicity of CAR-T cell therapy [24,25]. In a retrospective analysis among 228 patients with B cell-derived non-Hodgkin lymphoma and acute lymphoblastic leukemia treated with anti-CD19 CAR-T, the investigators identified that the antibiotic-exposed group (piperacillin/tazobactam, meropenem, and imipenem/cilastatin) showed poor overall survival (OS) associated with a higher rate of ICANS and CRS compared to those unexposed to antibiotics [13]. This study suggests that the microbiome alteration by antibiotics is the key factor for modulating the anticancer activity of CAR-T cells. Notably, the authors further evaluated the association between baseline microbiome and CAR-T therapy’s clinical response in patients without antibiotic exposure. The authors discovered that the increased relative abundance of specific bacterial taxa, such as *Ruminococcus*, *Faecalibacterium*, and *Bacteroides*, was associated with a better therapeutic response and less toxicity [13].

In a separate study, Stein-Thoeringer et al. evaluated the possible influence of various antibiotics on the anticancer activity of CAR-T cells [26]. The author noted that meropenem, cefepime, ceftazidime, and piperacillin-tazobactam were high-risk antibiotics that significantly impaired the response to CAR-T cell therapy. Mechanistically, researchers discovered that the abundance of *Bifidobacterium longum* was significantly increased in the fecal samples of patients who did not receive high-risk antibiotics. Moreover, increased *Bifidobacterium longum* is associated with the prolonged survival of patients after undergoing CAR-T cell therapy. This indicates that *Bifidobacterium longum* contributes to the long-term survival outcomes of CAR-T cell therapy.

Interestingly, vancomycin may improve the antitumor activity of CAR-T cells, while treatment using some antibiotics negatively influences CAR-T activity [25,27]. Uribe-Herranz et al. reported that inhibition of tumor growth by CAR-T was significantly enhanced with vancomycin co-administration [27]. Vancomycin treatment improved tumor-associated antigens (TAAs) cross-presentation in dendritic cells, enhancing tumor eradication. Moreover, the author noted that CD8 T cells and expression of cytotoxic associated genes (granzyme B, perforin, and IFNγ) were dramatically increased in mice treated with CAR-T cells and vancomycin. The gut microbiome composition of the mice treated with vancomycin was analyzed to understand the potential effect of vancomycin on the microbiome. Authors have identified that vancomycin treatment dramatically decreased alpha diversity and increased vancomycin-resistant bacteria, including *Enterobacteriaceae* and *Sutterelaceae*.

The microbiome influences the clinical response and adverse effects of anti-B-cell maturation antigen CAR-T cell therapy, in addition to the anti-CD19 CAR-T cell therapy. For example, Hu et al. evaluated differential reactivity and CRS grade by microorganism type for BCMA-CAR T therapy [28]. *Faecalibacterium*, *Roseburia*, and *Ruminococcus* were more increased in the complete remission (CR) group. *Prevotella*, *Collinsella*, and *Bifidobacterium* were detected with higher frequencies in the partial response (PR) group. The level of *Bifidobacterium* regarding toxicity in the severe CRS group rose gradually from the onset to the period of CRS. In contrast, *Leuconostoc* showed consistently higher levels during severe CRS. This revealed that the severity of CRS by CAR-T cell therapy can be predictable by monitoring microbiome composition in the intestinal microbiome, thereby optimizing more patient management.

Overall, the pre-clinical and clinical studies above suggest that the microbiome is a critical factor in modulating CAR-T activity and toxicity (Table 1). However, there is a need for further studies to fully understand the relationship between CAR-T therapy and the microbiome.

### 2.2. Mesenchymal Stem Cell (MSC) and MSC-Derived Exosome

MSCs are self-renewing adult stem cells with multi-lineage differentiation potential obtained from diverse tissues comprising bone marrow, amniotic membrane, umbilical cord, and adipose tissue [34,35]. MSCs have diverse abilities to modulate immune response.

MSC expresses T cell immunoreceptors with Ig and ITIM domains (TIGIT), which are immune checkpoints [36]. TIGIT was noted to induce immune suppression by binding to CD155 on T cells, NK cells, and APCs. Specifically, TIGIT suppresses T cell proliferation and differentiates them into regulatory T cells. Treg cells inhibit the secretion of inflammatory cytokines (Interferon Gamma [IFN-γ] and Tumor Necrosis Factor Alpha [TNF-α]) from Th1 cells [37]. In addition, Treg cells regulate inflammation by secreting anti-inflammatory cytokines (e.g., TGF-β, IL-10).

Interestingly, N. Borcherding et al. revealed that MSCs can modulate immune responses through mitochondrial transfer to T cells [29]. Mitochondria transferred from MSCs to T cells promote differentiation into Treg cells, decreasing Th17 cells. The increase in Treg cells ultimately leads to immune suppression through the secretion of anti-inflammatory cytokines by Treg cells and the inhibition of inflammatory cytokine secretion by Th1 cells. MSCs aid in treating various inflammatory diseases using these immunosuppressive properties [38]. In addition, it was observed that treating anti-inflammatory diseases using MSCs causes changes in the patient’s microbiome or in experimental animals. These changes in the microbiome induced by MSCs are closely associated with disease improvement.

For example, administering human umbilical mesenchymal stem cells (hUMSCs) for treating RA can alter the diversity of the intestinal microbiome [39]. Li et al. identified that *Clostridiaceae* and the *Clostridium* genus were prevalent in the mouse liver, along with collagen-induced arthritis (CIA). In contrast, the abundance of *Lactococcus* and *Bacteroides* decreased. Disease-associated microbiomes such as *Clostridiaceae* and the *Clostridium* genus were reduced when hUMSC was treated in mice with CIA, whereas the frequency of *Lactococcus* and *Bacteroides* was restored to normal. Furthermore, the proportions of Epsilonproteobacteria, *Campylobacterales*, *Bacteroidaceae*, and *Helicobacteraceae* were dramatically increased. These microbes synthesized their derivative metabolites, including indole, indoleacetic acid (IAA), and indole-3-lactic acid (ILA), known as ligands for the Aryl hydrocarbon receptor (Ahr) that can control the stability of tight junctions in the gut [30,40]. Activated Nuclear Factor Kappa-Light-Chain-Enhancer of Activated B Cells (NF-Kb) p65 in tight junction by IFN-γ and TNF-α in inflammatory conditions can reduce the expression of tight junction proteins (Occludin and Zonula occludens-1 [ZO-1]). Therefore, the intestinal barrier destabilizes, allowing harmful microbiomes to invade the gut. However, upon activation of Ahr by microbiome-derived metabolites, the expression of ZO-1 and Occludin is restored by inhibiting the activity of NF-Kb p65 that induces stability of the tight junction, resulting in the enhancement of intestinal barrier function.

In addition, gut microbiome composition alterations were observed in mice with hypoxia-induced Pulmonary Hypertension (PH) [41]. The researcher noted an increase in both Firmicutes and Melainabacteria in the feces of these mice, whereas Bacteroidetes and Proteobacteria decreased. Increased intestinal Firmicutes can induce inflammatory responses by triggering innate immune responses through Firmicutes-derived peptidoglycan [42,43]. Conversely, intestinal Bacteroidetes are essential in maintaining immune suppressive activity by inducing Treg cell activation to secrete anti-inflammatory IL-10 through Bacteroidetes-derived polysaccharides [31,43]. Therefore, decreasing Bacteroidetes in mice with hypoxia-induced PH may cause further intestinal inflammation. Interestingly, treatment with MSCs in mice with PH restored the composition of Bacteroidetes and Proteobacteria, while reducing Firmicutes and Melainabacteria, which correlated with alleviated inflammation and progression of PH [41]. While the precise mechanisms remain unclear, it is hypothesized that MSCs modulate the gut microbiome through their immunomodulatory properties, which may promote an anti-inflammatory environment and foster the growth of beneficial microbial populations. Future studies are needed to investigate whether MSCs can similarly restore microbiome composition in other inflammatory diseases and how these alterations contribute to therapeutic outcomes.

Furthermore, MSC-derived exosomes influenced the gut microbiome in inflammatory diseases [44,45,46]. MSC-derived exosomes are extracellular vesicles of MSC origin containing multiple active biomolecules, including proteins, nucleic acids (mitochondrial DNA, messenger RNA (mRNA), and microRNA), and lipids [32]. These active biomolecules are essential in regulating inflammation. For example, the anti-inflammatory compound (miR-181a) enclosed by MSC-derived exosomes could suppress pro-inflammatory cytokines released by natural killer, dendritic, and T cells [47]. MSC-derived exosomes can be considered great candidates for therapeutic agents in various inflammatory diseases considering the importance of MSC-derived exosomes in modulating gut microbiome and inflammation.

L Gu et al. proved that treating MSC-derived exosomes in ulcerative colitis (UC) changes gut microbiome composition, thereby alleviating UC progression [33]. MSC-derived exosome treatment enriched a beneficial microbiome such as *Lactobacillus* and depleted a pathogenic bacteria like *Bacteroides* in the UC mouse model induced by dextran sodium sulfate. Furthermore, suppressed pro-inflammatory cytokines (e.g., TNF-α) in adipocytes were decreased by exosomal miR-181a in MSC-derived exosomes. It also enhanced intestinal barrier integrity by enhancing the expression of tight junction proteins (e.g., Claudin-1 and ZO-1) in human colonic epithelial cells. These results imply that MSC-derived exosomes exert direct anti-inflammatory effects and influence intestinal microbiome balance.

In addition, the therapeutic application of MSC-derived exosomes is assessed in liver damage [45]. Yi et al. found that MSC-derived exosomes reduced inflammation and necrosis in the liver tissue in a mouse model of liver damage with inflammatory infiltration and necrosis by altering the frequency of pro- and anti-inflammatory macrophage [45]. These macrophage changes are associated with altering gut microbiome composition, with enriched beneficial bacterial species (*Faecalibaculum*) and depleted pathogenic strains (*Intestinimonas*). *Faecalibaculum* increased by MSC-derived exosomes is positively correlated with ascorbic acid, which is an antioxidant agent important in controlling immune responses. Moreover, these changes in the microbiome balance were associated with increased liver enzymes such as alanine aminotransferase (ALT), aspartate aminotransferase (AST), and albumin in the serum of MSC-derived exosome-treated mice. ALT and AST are essential in the liver’s energy production and amino acid metabolism [48]. Furthermore, albumin is a major protein component in the blood and performs various functions, including maintaining serum osmotic pressure and transporting substances in the blood. The recovery of these liver enzymes and albumin through MSC-derived exosomes highlights their strong therapeutic potential in treating inflammatory disease.

Sjögren’s syndrome (SS) is another disease that may be treated by MSC and MSC-derived exosomes [49]. SS is an autoimmune disease that induces inflammation and damage primarily to the exocrine glands [50]. This condition causes the immune system to damage cells and tissues in the exocrine gland, thereby decreasing the function of the salivary and lacrimal glands. Zou et al. revealed the effects of the hUMSCs and hUMSCs-derived exosomes on SS [49]. Mice suffering from SS injected with hUMSCs or hUMSCs-derived exosomes showed that the abundance of pro-inflammatory bacteria, such as *Escherichia-Shigella* and *Enterorhabdus*, was significantly reduced compared to WT mice. In contrast, the abundance of beneficial bacteria, like the *Eubacterium xylanophilum* group, increased. Furthermore, treatment with hUMSCs or hUMSCs-derived exosomes resulted in an increased and decreased proportion of Treg and Th17 cells, respectively, accompanied by alterations in cytokine levels, including the downregulation of pro-inflammatory cytokines (IL-6 and IFN-γ) and the upregulation of anti-inflammatory cytokines (TGF-β1 and IL-10). In summary, MSC or MSC-derived exosomes are a viable therapeutic option for treating inflammatory disease (Table 1).

### 2.3. FDA-Approved Antibody

In biopharmaceuticals, antibody-based therapy is the main therapeutic method for treating various diseases. Notably, ICIs that modulated inhibitory signals in T cells revolutionized antibody-based cancer therapy. Keytruda, a phosphodiesterase-1 inhibitor, dramatically improved the OS of the late stage of relapse and/or patients with refractory melanoma [51]. Unfortunately, this strong effect of ICI varies among patients, therefore, multiple studies were conducted to understand the possible factors contributing to the therapeutic effect of ICI. Intriguingly, some evidence has revealed that the antitumor effect of ICIs is closely related to a patient’s microbiome composition and probiotics usage [52,53,54,55]. The association between microbiome and ICI has been thoroughly reviewed in many previous articles; therefore, in this study, we want to show new possibilities of the role of microbiome in different antibody therapies for treating cancer, inflammatory, and infectious diseases (Table 2). However, there are significant gaps in our understanding of the specific roles of the microbiome due to the complexity of investigating the mode of action of the microbiome in each antibody therapy. Therefore, this review will primarily concentrate on the alterations in and associations between the microbiome and the therapeutic effects of antibodies, to provide novel insights for researchers in this field.

First, adalimumab is an anti-TNFα antibody that is primarily used in treating various inflammatory diseases [63]. In Crohn’s disease (CD), treatment with adalimumab altered the microbiome’s composition by decreasing Proteobacteria and elevating Firmicutes, Bacteroidetes of the family *Lachnospiraceae* in pediatric patients [64]. Reduced Proteobacteria is associated with decreased Th17 cells, which may lead to reduced secretion of inflammatory cytokines (e.g., TNF-α, IL-6) [65]. The family *Lachnospiraceae* produces butyrate, which promotes the differentiation of Treg cells in the colon [56,57]. Subsequently, IL-10 secreted by these Treg cells suppresses inflammatory responses in mucosal tissues, including the colon. Therefore, reducing Proteobacteria and increasing *Lachnospiraceae* may help reduce inflammation in inflammatory conditions.

Infliximab (IFX), another anti-TNFα antibody, is a therapeutic agent for patients with CD [66]. The *Lactobacillus fermentum* count decreased following treatment with IFX in patients with CD, indicating the pro-inflammatory activity of *Lactobacillus fermentum*. In contrast, *Bacteroides fragilis*, with a crucial anti-inflammatory role involving suppressing inflammation, showed no significant difference in its count pre- and post-treatment. In addition, patients with CD with an increased number of *Clostridiales*, known as defensive commensals, exhibited an improved response to IFX treatment than those with lower levels.

Bevacizumab, a monoclonal antibody against human vascular endothelial growth factor (VEGF), has been proven to influence the gut microbiome in patients with metastatic colorectal cancer (mCRC) [67]. The microbial diversity of the feces samples in the Bevacizumab group was markedly higher than those in the non-Bevacizumab group, indicating that high bacterial diversity may positively impact the treatment of mCRC [67]. Particularly, the levels of *Bifidobacterium* and *Lactobacillus* species were increased in the Bevacizumab-receiving group. *Lactobacillus* and *Bifidobacterium* exerted anticancer effects by inducing apoptosis through the production of antitumor proteins and cell cycle regulation [58]. Faghfoori et al. demonstrated that *Bifidobacterium* upregulated the expression of pro-apoptotic genes (cysteine-aspartic acid protease 8, Fas-Receptor, Bcl-2 associated death promoter) in CRC cell lines [58]. Similarly, Tiptiri-Kourpeti et al. discovered that *Lactobacillus* increased the mRNA expression levels of TNF-related apoptosis-inducing ligand, which activates external apoptotic signaling pathways, by 60-fold in CRC cell lines [59]. Furthermore, *Lactobacillus* was associated with a 10-fold lower mRNA expression of cyclin D1, which is required for promoting the G1 phase of the cell cycle, and the anti-apoptotic protein Survivin, thereby suggesting that *Lactobacillus* and *Bifidobacterium* may contribute to anticancer efficacy.

Denosumab, an immune-modulating drug and a human monoclonal antibody against Receptor Activator of Nuclear factor Kappa-B Ligand (RANKL) was effective in alleviating the dinitrobenzene sulfonic acid-induced colitis by immune activation and modulation of gut microbiome composition [60]. Denosumab significantly decreased inflammatory cytokine expression (TNF-α, IL-6, IL-1β) in the colonic mucosa of mice, indicating an immune reduction upon RANKL blockade. Furthermore, it enhanced the enrichment of beneficial bacteria such as Firmicutes and *Clostridiales*. *Clostridiales* is an energy source for the intestinal epithelium and decreases intestinal pH, thereby enhancing protection against pathogenic bacteria.

Dupilumab, an anti-interleukin-4 receptor antibody, is a monoclonal antibody effective in managing atopic dermatitis (AD). Dupilumab treatment demonstrated reduced skin inflammation and restored epidermal barrier function by blocking Interleukin-4 Receptor Alpha [68]. In addition, dupilumab decreased the proportion of *Staphylococcus* (*S*). *aureus* genera and increased *S. epidermidis* and *S. hominis* in patients with AD. Notably, *S. aureus* expresses virulence factors that worsen the disease. Contrary, commensal *staphylococcal* species, including *S. epidermidis* and *S. hominis*, restrict the colonization of the pathogen together.

The association between treatment with Ustekinumab (UST), an antibody a targeting the common p40 subunit of IL-12 and IL-23, and the gut microbiome can be a novel biomarker for predicting treatment response in patients with CD [69]. The study revealed an increased relative abundance of *Faecalibacterium* in patients upon achieving remission at 6 weeks after UST treatment, indicating a possible beneficial effect on CD pathogenesis. *Faecalibacterium* induced the development of Treg cells in the colon, thereby increasing the anti-inflammatory cytokine IL-10 levels [61]. In addition, butyrate produced by *Faecalibacterium* inhibits NF-κB activity in intestinal epithelial cells, which blocks the production of pro-inflammatory cytokines such as IL-8. Furthermore, butyrate suppresses the inflammatory Wnt/c-Jun N-terminal kinase (JNK) signaling pathway by inhibiting Histone deacetylase (HDAC), ultimately reducing IL-8 production. However, it is uncertain whether these effects are solely attributable to the butyrate produced by *Faecalibacterium*.

*Clostridium difficile* infection (CDI) is caused by the toxins of *C. difficile*, Toxin A (TcdA), and TcdB [62]. TcdA and TcdB inactivate the Rho GTPase family within colonic epithelial cells, leading to inflammation. Bezlotoxumab, an antibody against TcdB, effectively eliminates toxins from the feces and significantly reduces the recurrence rate in patients with CDI [70]. Bezlotoxumab treats the infection and induces changes in the gut microbiome composition. Bezlotoxumab treatment increases Firmicutes while decreasing Bacteroidetes and Proteobacteria. Proteobacteria reduction led to a decrease in Th17 cells and suppression of inflammatory cytokine secretion, whereas the increase in Firmicutes promoted Treg cell differentiation, mitigating inflammatory responses [70].

The therapeutic effect of combining an anti-EGFR antibody (cetuximab) with an anti-PD-L1 antibody (avelumab) varies in patients with mCRC and non-small cell lung cancer (NSCLC) depending on specific microbiome compositions [71]. In patients who responded to the combination therapy with cetuximab and avelumab, two butyrate-producing bacterial species, *Agathobacter* and *Blautia*, were enriched. Notably, a higher population of *Agathobacter* and *Blautia* were associated with better progression-free survival, indicating that the abundance of butyrate produced by *Agathobacter* and *Blautia* may be responsible for enhancing antitumor activity. However, the precise mechanism by which butyrate contributes to increased antitumor activity remains unclear, requiring further research.

## 3. Enhancing Immunotherapy Efficacy Through Gut Microbiome, Probiotics, and Metabolites

Interest in probiotics and microbiome-derived metabolites has increased with the rapid advancement of research on the human microbiome. Probiotics, which are living microorganisms that offer health benefits to the host, have been shown to have various effects, including improving gut health and regulating the immune system [72]. Most beneficial effects mediated from probiotics come through their metabolites. Metabolites, as products of microbial metabolic processes, include short-chain fatty acids (SCFA) that significantly impact human physiological functions [73,74]. SCFAs improve insulin sensitivity in peripheral tissues to prevent metabolic disorders like type 2 diabetes. Moreover, it regulates immune cell functions, such as promoting Treg differentiation to reduce inflammatory responses. In addition, butyrate, a type of SCFA, acts as an HDAC inhibitor, contributing to the prevention of cancer and inflammation [73]. We aimed to discuss the impact of probiotics and their metabolites on the efficacy of immunotherapy, considering the importance of the microbiome and its metabolites in modulating the immune response.

First, researchers investigated the potential role of the microbiome and probiotics on the efficacy of cancer vaccines. Cancer vaccines consist of defined cancer antigens that stimulate the immune system by targeting cancer cells for the treatment or prevention of cancer [74]. According to Jing et al., *Lactobacillus rhamnosus GG* (LGG) combined with a probiotic, jujube powder, markedly enhances the therapeutic effect of cancer vaccines by regulating gut microbiome and lipid metabolism [75]. They reported that using LGG increased the relative abundance of *Muribaculaceae*, which improved antitumor activity and decreased microbial α-diversity in mice. Similarly, Wen et al. investigated the role of specific microbiomes in the effectiveness of the rotavirus vaccine, specifically focusing on LGG [76]. They discovered a rise in rotavirus antigens and viral titers in cells treated with LGG. Furthermore, the previous high-dose administration of LGG in pigs vaccinated with the attenuated human rotavirus vaccine enhanced the virus-specific IFN-γ T cell response. This result demonstrated the utilization of probiotics for enhancing vaccine efficacy. In addition, Lione et al. found that controlling specific microbiomes is important for maximizing the therapeutic effect of neoantigen cancer vaccine (NCV) [77]. They identified that a decrease in *Lachnospiraceae* can improve immune response using NCV by inhibiting Treg’s function. Moreover, they further demonstrated that a strong immune response mediated by NCV was observed with an increased frequency of *Bacillus*, *Thermoactinomyces*, and *Bosea*, suggesting that they could be considered a great combination with NCV for better cancer vaccine development.

*Lactobacillus species*. (*L. acidophilus*, *L. reuteri*, *L. salivarius*) was significant in modulating the immune response in chickens, particularly affecting antibody responses toward Newcastle Disease Virus vaccine [78]. *Lactobacillus*, a non-pathogenic Gram-positive bacterium residing in the gut microbiome of animals, is a probiotic. It was discovered that when *L. salivarius* was administered to the chicken, it dampened IFN-γ production and promoted Th2 immune responses. These immune responses caused an increased serum antibody production. In addition, administration of L. acidophilus induced the expression of Th1-associated cytokines, including IFN-γ and IL-12, leading to elevated levels of IgG and IgM antibodies targeting KLH. This indicated that specific treatments with *Lactobacillus* can stimulate antibody production and enhance immune cell activation through different mechanisms. Finally, Hwang et al. found that butyrate produced by the microbiome, including *Muribaculaceae*, *Mucispirillum*, and *Ruminoccaaceae*, raised antibodies and anti-inflammatory cytokines production, further increasing the immunogenicity of mucosal vaccines against severe acute respiratory syndrome coronavirus 2 [79].

Metabolites affect the efficacy of vaccines and may alter the efficacy of CAR-T cell therapy in cancer treatment. Luu et al. indicated that SCFAs, pentanoate, and butyrate increased the anticancer activity of cytotoxic T lymphocytes (CTL) and CAR-T cells through metabolic and epigenetic reprogramming [80]. It was shown that SCFAs promote the production of effector cytokines like TNF-α and IFN-γ by CTLs and increase the mechanistic target of Rapamycin activity in CTLs. Furthermore, pre-treating pentanoate with CAR-T cells increased cell proliferation and IL-2 secretion in vitro. Notably, when pentanoate or butyrate pre-treated OT-I CTLs were transferred into mice engrafted with the B16-OVA melanoma cell line, tumor growth was significantly delayed compared to the control. Pentanoate pre-treated CAR T cells in different tumor models, such as pancreatic cancer, reduced tumor volume and increased IFN-γ and TNF-α production in tumor microenvironment. This indicates that pentanoate treatment enhances the antitumor function in CAR-T cells both in vitro and in vivo.

Similarly, Luo’s research team discovered that gallic acid (GA) significantly promotes CD19 CAR-T efficacy. They confirmed that GA treatment increased IL-4 expression and Janus kinase 3-signal transducer and activator of transcription 3 activation in CAR-T cells, causing enhanced cytokine production and CAR-T expansion both in vitro and in vivo [81]. These findings indicate that the pre-clinical response of CAR-T therapy, when combined with GA, can substantially improve lymphoma. In summary, combining microbiomes, probiotics, and metabolites is a viable strategy for optimizing and accelerating immunotherapy activity. To date, we have investigated the roles and correlations of probiotics and metabolites in immunotherapy. These findings are synthesized and summarized in Figure 2.

## 4. Revolution of Microbiome Technologies: Tailored Microbiome by Genetic Engineering

Various novel approaches utilizing microbiomes have been actively evaluated due to new evidence highlighting the gut microbiome’s significant role in modulating the efficacy of immunotherapy. While there are various approaches to using microbiomes to directly modulate disease, exploiting the microbiome as a drug delivery platform can offer significant benefits in overcoming the limitations of conventional treatments. For example, administering therapeutic reagents via microbiome can potentiate the effect of the small recombinant protein, whose therapeutic duration is limited due to its short half-life. Wei et al. developed a novel delivery system for α-melanocyte-stimulating hormone (α-MSH) with *Bifidobacterium* for inflammatory bowel disease (IBD) therapy [82]. The precise pathogenesis of IBD is uncertain; however, numerous studies have identified that inflammatory cytokines play a critical role in IBD progression [83]. Therefore, many anti-inflammatory drugs, including α-MSH that inhibits the NF-κB pathway and reduces the expression levels of various inflammatory cytokines (e.g., TNF-α, IL-6), have been exploited [84]. However, due to the short half-life of α-MSH in vivo, its overall effect is marginal. Researchers engineered *Bifidobacterium longum* to express α-MSH stably to prolong the therapeutic effect of α-MSH, and the administration of engineered Bifidobacterium longum resulted in an increase in α-MSH and a reduction in the expression of inflammatory cytokines (TNF-α, IL-6, IL-1β) [82].

In addition, Zhang et al. developed a *Bifidobacterium longum* strain carrying the abfA gene cluster to improve arabinan degradation for treating functional constipation (FC) [85]. FC is a chronic condition characterized by constipation without structural or organic disease and is prevalent worldwide. Arabinan, a non-digestible polysaccharide found in plant cell walls, was fermented by specific gut microbes, during which the abfA cluster critically produced various metabolites (acetate, butyrate, uracil, and chenodeoxycholic acid). These metabolites increased intestinal water content and stool volume, alleviating constipation. A *Bifidobacterium longum* strain expressing the abfA gene cluster was generated to explore this therapeutic potential. Transplantation of abfA cluster-expressing microbiome into germ-free mice with FC effectively ameliorated their symptoms.

Gong et al. developed *Escherichia coli* Nissle 1917 (EcN), which can selectively deliver butyrate to the intestinal mucosa for chronic colitis treatment [86]. Butyrate acts as an inhibitor of Histone Deacetylase 3, blocking inflammatory signaling pathways. In addition, it functions as an inhibitor of Gasdermin D, a key player in pyroptosis (i.e., inflammatory cell death), thereby contributing to maintaining intestinal mucosal homeostasis. Using these mechanisms, researchers introduced a butyrate synthesis pathway into EcN with selective anaerobic symbiotic bacteria, confirming its efficacy in reducing the risk of colitis. Similarly, Dosoky et al. developed EcN overexpressing N-acyl-phosphatidylethanolamine (NAPE) for obesity [87]. They proved, for a series of bioactive lipids, that NAPE is an anti-obesity compound and discovered that the administration of EcN overexpressing NAPE was resistant to diet-induced obesity caused by a high-fat diet.

Furthermore, engineered microbiomes can be used for the prevention and treatment of viral infections. Vangelista et al. used vaginal *Lactobacillus jensenii* to produce anti-HIV-1 proteins (C1C5 Regulated upon Activation, Normal T cell Expressed and Secreted [RANTES]) for HIV-1 infection prevention [88]. CCR5 is a major co-receptor used by HIV-1 to facilitate viral entry into CD4+ T cells, macrophages, and dendritic cells with the CD4 receptor. C1C5 RANTES is a modified chemokine that acts as a CCR5 ligand, blocking HIV-1 from accessing the receptor by serving as a CCR5 antagonist. C1C5 RANTES, unlike conventional RANTES, developed by Vangelista, effectively inhibits HIV-1 infection without inducing inflammatory responses. Treatment of vaginal *Lactobacillus jensenii* expressing C1C5 RANTES prevents HIV infection in monkeys.

Finally, genetically modified microbiomes are utilized in cancer therapy. For example, Vincent et al. engineered a non-pathogenic EcN strain to secrete an adaptor consisting of a super folder GFP (sfGFP) domain that binds to GFP-specific CARs (i.e., ProCAR) and a heparin-binding domain (HBD) from placental growth factor 2, which attaches to the extracellular matrix (ECM) components such as collagen, fibronectin, and heparan sulfate proteoglycans (HSPGs) within tumors [89]. The secreted adaptor binds to ECM components, and the sfGFP domain of the tag protein induces ProCAR T cell activation. Consequently, ProCAR T cells remain inactive outside the tumor, reducing off-target immune responses and minimizing side effects, while overcoming the limitations of conventional CAR-T therapy in solid tumors. Overall, engineered microbiomes offer innovative therapeutic solutions across diverse medical conditions.

## 5. Conclusions

In conclusion, exploring the interaction between the gut microbiome and various immunotherapies, including CAR-T cells, MSC, and antibody-based drugs, is an emerging field with significant potential for enhancing therapeutic outcomes in various diseases. Current research has highlighted that gut microbiome composition and activity can significantly impact the efficacy and immune responses associated with these treatments. Furthermore, advances in engineered microbiome research emphasize the potential to harness genetically modified microbes as delivery platforms for therapeutic agents or as adjuvants to optimize vaccine efficacy. These engineered systems provide innovative strategies to amplify therapeutic outcomes by targeting specific mechanisms or pathways that enhance immune function.

The underlying mechanisms governing microbiome–immunotherapy interactions are incomplete and require further investigation despite these advancements. Understanding these complex interactions is crucial for developing targeted probiotic or microbiome-based strategies to optimize treatment effectiveness and clinical outcomes. The relevance of microbiomes in immunotherapy is continually emphasized by ongoing clinical evaluations continually reinforcing their potential as a key component in next-generation therapeutic development (Appendix A). Microbiome modulation could unlock innovative advances in immunotherapy with further exploration and refinement, laying the foundation for more effective and personalized treatments.

## Figures and Tables

**Figure 1 ijms-26-00856-f001:**
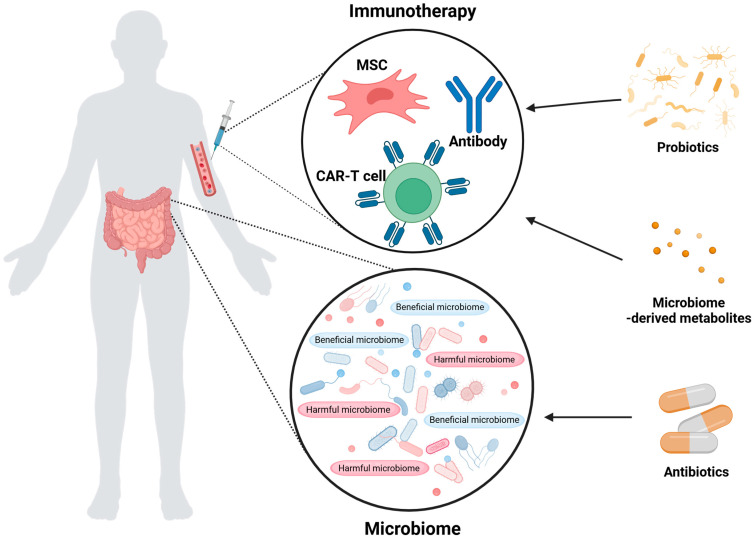
Immunotherapy–gut microbiome interaction. The efficacy of immunotherapies including CAR-T cells, MSCs, and antibodies, is strongly modulated by the composition of the patient’s gut microbiome. Antibiotics administered prior to immunotherapy can modulate the microbiome and influence which probiotics and metabolites are present.

**Figure 2 ijms-26-00856-f002:**
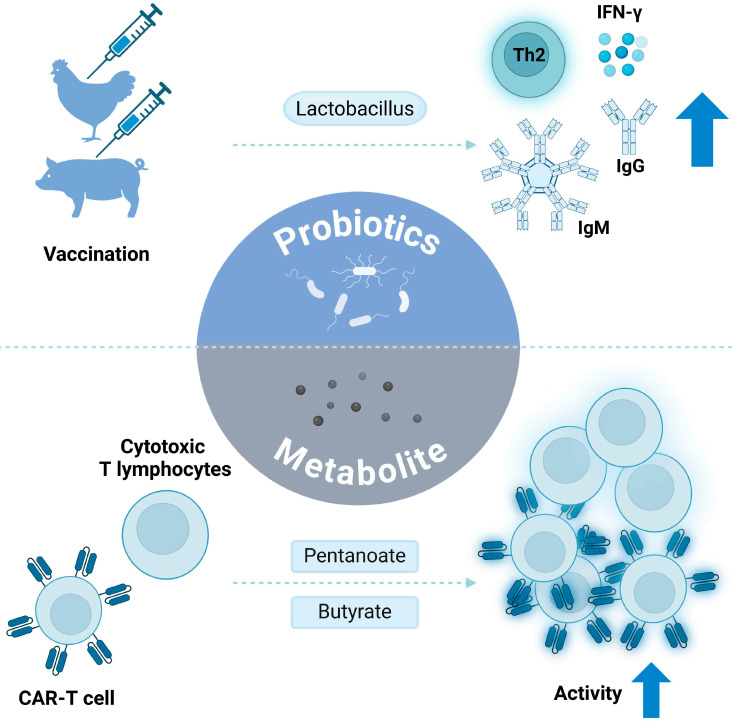
Use of microbiome-derived metabolites and probiotics: probiotics and microbiome-derived metabolites impact the efficacy of antigen-based vaccines, CAR-T cells, and cytotoxic T lymphocytes.

**Table 1 ijms-26-00856-t001:** Impact of the microbiome on the efficacy of cell-based immunotherapy (CAR-T cell and mesenchymal stem cell).

Immunotherapy	Disease	Factor Influencing Gut Microbiome	Microbial Species	Immunotherapy Efficacy Difference	Ref.
CD19 CAR-T	B cell lymphoma	Vancomycin	↑: *Enterobacteriaceae*, *Sutterellaceae*	Increased antigen presentation, CD8+ T cell increase, and enhanced T cell activation	[22,25]
			↓: *Ruminococcaceae*, *Lachnospiraceae*		
	NHL, ALL	PIM (piperacillin/tazobactam, imipenem/cilastatin, and meropenem)	↑: *Ruminococcus*, *Faecalibacterium*, *Ruminococcaceae*, *Faecalibacterium prausnitzii*, *Ruminococcus bromii*	High responsiveness and lack of toxicity to CAR-T	[23]
	Refractory and relapsing lymphoma	-	↑: *Bifidobacterium longum*	Longer survival after CAR-T cell therapy	[24]
BCMA CAR-T	MM	-	CR: *Faecalibacterium*, *Roseburia*, *Ruminococcus*	CAR-T responsiveness and CRS grading vary based on patients’ gut microbiome composition	[26]
			PR: *Prevotella*, *Collinsella*, *Bifidobacterium*		
			Severe CRS: *Bifidobacterium*, *Leuconostoc*		
HUMSC	Collagen-induced arthritis (CIA)	HUMSC	↑: *Epsilonproteobacteria*, *Campylobacterales*, *Bacteroidaceae*, *Helicobacteraceae*	Enhanced mucosal immune function	[29]
	Pulmonary hypertension (PH)	HUMSC	↑: *Bacteroidetes*, *Proteobacteria*, *Bacteroidaceae*, *Prevotellaceae*, *Tannerellaceae*, *Lachnospiraceae*	Mitigation of gut microbiome imbalance, reduced inflammation	[30]
			↓: *Firmicutes*, *Melainabacteria*		
MSC-exosome	Liver damage	MSC-exosome	↑: *Faecalibaculum*	Improved liver enzyme levels, albumin, coagulation factors, and reduced liver inflammation	[31]
			↓: *Intestinimonas*		
	Ulcerative colitis (UC)	MSC-exosome	↑: *Lactobacillus*	Improved gut microbiome environment, gut barrier function maintenance, reduced inflammation, enhanced cell survival	[32]
			↓: *Bacteroides*		
	Sjogren’s syndrome(SS)	MSC-exosome	↑: *Eubacterium xylanophilum_group*	Reduced inflammatory cytokines, improved inflammatory status, and better Treg/Th17 balance	[33]
			↓: *Escherichia-Shigella*, *Enterorhabdus*		

↑: Indicates an increase in the abundance of the microbial species, which correlates with the described effects on immunotherapy efficacy. ↓: Indicates a decrease in the abundance of the microbial species, which correlates with the described effects on immunotherapy efficacy.

**Table 2 ijms-26-00856-t002:** Impact of the microbiome on the efficacy of antibody-based immunotherapy.

Antibody	Target	Disease	Microbial Species	Effect	Ref.
Adalimumab	TNFa	Crohn’s disease (CD)	↑: *Lachnospiraceae*	Reduced intestinal inflammation	[55]
			↓: *Proteobacteria*		
Infliximab	TNFa	Crohn’s disease (CD)	↑: *Clostridiales*	Reduced inflammation	[56]
			↓: *Lactobacillus fermentum*		
Bevacizumab	Human vascular endothelial growth factor (VEGF)	Metastatic colorectal cancer (mCRC)	↑: *Bifidobacterium*, *Lactobacillus*	Increased anticancer efficacy	[57]
			↓: *F. nucleatum*		
Denosumab	RANKL (Receptor Activator of Nuclear Factor Kappa-Β Ligand)	Dinitrobenzo sulfonic acid (DNBS) induced colitis	↑: *Firmicutes*, *Clostridiales*	Decreased inflammatory cytokines, reduced immune response	[58]
Dupilumab	IL-4, IL-13	Atopic dermatitis (AD	↑: *S. hominis*, *S. epidermidis*	-	[59]
			↓: *Staphylococcus*, *S. aureus*		
Ustekinumab	IL-12/IL-23	Crohn’s disease (CD)	↑: *Faecalibacterium*	Reduced inflammation	[60]
Bezlotoxumab	Toxin TcdB	Clostridium difficile infection (CDI)	↑: *Firmicutes*	Reduced inflammation and toxicity	[61]
			↓: *Bacteroidetes*, *Proteobacteria*		
Cetuximab	EGFR	Metastatic colorectal cancer (mCRC)	*Agathobacter M104/1*, *Blautia SR1/5*	Regulation of immune response, enhanced T cell infiltration, increased anticancer efficacy	[62]

↑: Indicates an increase in the abundance of the microbial species, which correlates with the described effects on immunotherapy efficacy. ↓: Indicates a decrease in the abundance of the microbial species, which correlates with the described effects on immunotherapy efficacy.

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
