# Peer review of "Hidden Partner of Immunity: Microbiome as an Innovative Companion in Immunotherapy"

_ijms, 2025, doi:10.3390/ijms26020856_

Round 1

Reviewer 1 Report

Comments and Suggestions for Authors

In order to understand the association between immunotherapy efficacy and the microbiome, the authors have summarized the reported literatures in studying the relationship of microbiome with different forms of immunotherapies. The novelty of this review is that the authors have brought new knowledge in how to take advantage of the microbiome change with better therapeutic effects of antibodies in treating cancer or other diseases. Overall, I found this review is written in a good structure and is interesting to the readers. Before published, I have some comments below to help improve the quality of this review.

  1. There are grammar issues in the Abstract, please rewrite the sentences of “Therefore, understanding the potential factors that can modulate the effects of immunotherapy to improve its clinical outcome is critical. The microbiome, a living organism in the human body, has been historically identified as important in regulating a hostʹs immune response.” I suggest to rewrite the Abstract, put the summary of previous study at the beginning to replace the current first sentence. Keep the rephrased sentences with correct grammar as it is. At the end, elaborate more of what the authors have included in this review and state the novelty of the review.
  2. Introduction has described the background study very well. Please add the cited literatures at the end of “Scher et al. identified the enrichment of Prevotella copri in patients with new-onset RA…”; “Zegarra-Ruiz et al. discovered that dietary supplementation with…”; “Verizon et al. discovered an association between the effectiveness of Cytotoxic…”; “Sivan at al. found a correlation between the effectiveness of Programmed…” and “Smith et al. observed the efficacy and toxicity of Chimeric…”. 
  3. Part of Introduction: Please correct “In this study, we discovered that the low abundance…” changed it to the researcher discovered. At the end of Introduction, the authors have written “and antibody therapy excluding ICI, as numerous review articles have thoroughly discussed the association of the microbiome with ICI.” But from the authors’ summary of CTLA-4 and PD-L1 antibody therapy, only three literatures mentioned above. Probably need to add more citations here to demonstrated “numerous review articles”, or I would suggest change it to “as it is widely studied and well summarized of the microbiome associates with the effectiveness of ICI.”
  4. Part of 1-1. CART cell therapy: please make the definition of “CR group” and “PR group” when it showed first time when talking about “Hu et al. evaluated differential reactivity and CRS grade…”
  5. Part of 1-2. MSC and MSC-derived exosome: the authors have mentioned MSCs treatment restored the microbiome in the mice with hypoxia-induced PH. I am curious of how MSC could restore the composition of Bacteroidetes and Proteobacteria while reducing Firmicutes and Melainabacteria. What about in a different condition, is it possible that MSC could restore the microbiome that can alleviate inflammation in any diseases? Please let me know your thought. 
  6. Part of 2. Enhancing Immunotherapy Efficacy Through Gut Microbiome, Probiotic and Metabolites: for citation 79, the authors have made the conclusion as “These findings indicate that the clinical response of CAR-T therapy”. But it seems only the pre-clinical study according to the literature. Is there any clinical evidence of microbiome associate with CART efficacy? Please add it to the review if there is any clinical outcomes. 
  7. Figure 1: Change typo MCS to MSC; It is hard to tell what the beneficial microbiome is and what the harmful microbiome is because they are all in the same color, how about using the same color as light red and light blue to represent them?
  8. For the figures and table, there are no relevant description in the review. Can you please add them in the main text of the review accordingly?   

Reviewer 2 Report

Comments and Suggestions for Authors

In this review, the authors discussed the role of microbiome as an Innovative Companion in Immunotherapy. The authors discussed the following points: The effect of microbiome in immunotherapy (CAR-T cell therapy). Then the authors discussed enhancing immunotherapy through metabolites and probiotics, Afterthat, the authors discuseed tailored Microbiome by genetic engineering.

Unfortuntaley the idea is not novel, and previosuly discribed by other groups including the following mentioned below

such as

https://pmc.ncbi.nlm.nih.gov/articles/PMC10093606/

https://ehoonline.biomedcentral.com/articles/10.1186/s40164-023-00442-x

https://www.sciencedirect.com/science/article/pii/S2666379124001332

https://www.cell.com/cell-reports-medicine/fulltext/S2666-3791(24)00124-1

https://www.thelancet.com/journals/ebiom/article/PIIS2352-3964(22)00344-9/fulltext

Round 2

Reviewer 2 Report

Comments and Suggestions for Authors

No further comments